# Evaluation of the Potency of Anti-HIV and Anti-HCV Drugs to Inhibit P-Glycoprotein Mediated Efflux of Digoxin in Caco-2 Cell Line and Human Precision-Cut Intestinal Slices

**DOI:** 10.3390/ph15020242

**Published:** 2022-02-18

**Authors:** Martin Huličiak, Ivan Vokřál, Ondřej Holas, Ondřej Martinec, František Štaud, Lukáš Červený

**Affiliations:** 1Department of Pharmacology and Toxicology, Faculty of Pharmacy in Hradec Králové, Charles University, 50005 Hradec Králové, Czech Republic; hulicim@faf.cuni.cz (M.H.); francuk@gmail.com (O.M.); staud@faf.cuni.cz (F.Š.); cervenyl@faf.cuni.cz (L.Č.); 2Department of Pharmaceutical Technology, Faculty of Pharmacy in Hradec Králové, Charles University, 50005 Hradec Králové, Czech Republic; holao3aa@faf.cuni.cz

**Keywords:** drug–drug interactions, ABCB1, antiretrovirals, direct-acting antivirals, human precision-cut intestinal slices

## Abstract

The inhibition of P-glycoprotein (ABCB1) could lead to increased drug plasma concentrations and hence increase drug toxicity. The evaluation of a drug’s ability to inhibit ABCB1 is complicated by the presence of several transport-competent sites within the ABCB1 binding pocket, making it difficult to select appropriate substrates. Here, we investigate the capacity of antiretrovirals and direct-acting antivirals to inhibit the ABCB1-mediated intestinal efflux of [^3^H]-digoxin and compare it with our previous rhodamine123 study. At concentrations of up to 100 µM, asunaprevir, atazanavir, daclatasvir, darunavir, elbasvir, etravirine, grazoprevir, ledipasvir, lopinavir, rilpivirine, ritonavir, saquinavir, and velpatasvir inhibited [^3^H]-digoxin transport in Caco-2 cells and/or in precision-cut intestinal slices prepared from the human jejunum (hPCIS). However, abacavir, dolutegravir, maraviroc, sofosbuvir, tenofovir disoproxil fumarate, and zidovudine had no inhibitory effect. We thus found that most of the tested antivirals have a high potential to cause drug–drug interactions on intestinal ABCB1. Comparing the Caco-2 and hPCIS experimental models, we conclude that the Caco-2 transport assay is more sensitive, but the results obtained using hPCIS agree better with reported in vivo observations. More inhibitors were identified when using digoxin as the ABCB1 probe substrate than when using rhodamine123. However, both approaches had limitations, indicating that inhibitory potency should be tested with at least these two ABCB1 probes.

## 1. Introduction

Human immunodeficiency virus (HIV) and hepatitis C virus (HCV) infections are major global health problems. Over 100 million people are currently living with HIV or HCV [1,2,3], almost 30 million of whom have been prescribed a lifelong antiretroviral combination regimen or months of medication with combinations of direct-acting antivirals (DAA) [1,2]. Patients with HIV and/or HCV frequently have serious comorbidities that require the administration of additional pharmacotherapy [4,5,6,7,8,9,10], which increases the risk of drug–drug interactions (DDI) [11,12,13,14]. Although antivirals are highly effective and well tolerated, they share metabolic pathways with other drugs and reveal frequent interactions with membrane transporters. This creates the potential for pharmacokinetic DDI that could cause a victim drug’s plasma concentration to reach toxic or subtherapeutic levels [11,15]. Therefore, knowledge of the molecular mechanisms underpinning pharmacokinetic DDI is essential for selecting appropriate antivirals and optimal antiviral doses [11,15].

Most antivirals are thought to be substrates and/or inhibitors of P-glycoprotein (ABCB1) [16,17,18]. ABCB1 is an active efflux transporter that determines the disposition of many chemically, structurally, and functionally unrelated substances, and is considered to be a site of clinically relevant DDI [19]. Its polyspecificity is due to the presence of a large and flexible binding pocket containing several distinct transport-competent sites for rhodamine123 (RHD123), Hoechst 33342, digoxin, and prazosin [20,21,22]. ABCB1 localized in the apical membrane of enterocytes reduces the net intestinal absorption of orally administered drugs [19,20], mainly of compounds with low permeability that are minimally metabolized by cytochrome P450 [19,23,24,25,26,27,28,29,30]. DDI on intestinal ABCB1 are known to have clinical consequences: the inhibition of intestinal ABCB1 has been shown to increase the absorption of dabigatran, talinolol, fexofenadine, or digoxin [23,25,26,27,28], while ABCB1 induction reduces exposure to sofosbuvir and dabigatran [30,31]. It has been suggested that both antiretrovirals and DAA may inhibit intestinal ABCB1, but their activity in this respect has not been studied thoroughly.

Human-derived precision-cut intestinal slices (hPCIS) are miniature models of the intestine with a physiological 3D architecture that can be used to study the effects of intestinal metabolism and transporter activity on drug pharmacokinetics [32,33]. By conducting accumulation studies in hPCIS and measuring bidirectional transport across Caco-2 cell monolayers using RHD123 as a model transport substrate, we recently showed that atazanavir, lopinavir, maraviroc, ritonavir, saquinavir, ledipasvir, and daclatasvir inhibit ABCB1 in the intestine [34]. However, abacavir, tenofovir disoproxil fumarate (tenofovir DF), zidovudine, rilpivirine, etravirine, and sofosbuvir did not detectably inhibit RHD123 transport [34]. We used RHD123 as the ABCB1 probe in these studies because it was reported to be suitable for measuring ABCB1 inhibition in hPCIS [35] and cell models [21,36,37]. However, recent studies have shown that relying exclusively on RHD123 as the ABCB1 probe may prevent the detection of ABCB1 inhibitors that bind to other transporter-competent sites [20,21]. Therefore, complementary studies with probes that bind to other sites should be performed [20,21]. Here, we present the results of one such complementary study using the cardiac glycoside digoxin as the probe. Digoxin was suggested to bind to the large D site of ABCB1, which partially overlaps with the smaller RHD123 site [20,38], and its transport appears to be inhibited by a wider range of clinically relevant drugs than that of RHD123 [20]. In addition, multiple regulatory agencies list digoxin as a suitable ABCB1 substrate that can be used to test for clinical DDI [39,40]. The set of antivirals tested for ABCB1 inhibition using this probe included all of those used in our previous study [34], together with asunaprevir, darunavir, elbasvir, grazoprevir, and velpatasvir.

## 2. Results

### 2.1. Effect of Antiretrovirals and DAA on Bidirectional Transport of [^3^H]-Digoxin across Caco-2 Monolayers

We initially performed bidirectional transport experiments using [^3^H]-digoxin alone, for which the efflux ratio (rP_app_) was 9.53 ± 2.22. Adding the model ABCB1 inhibitor CP100356 monohydrochloride (CP100356) (2 µM) reduced the rP_app_ of [^3^H]-digoxin to 1.49 ± 0.11. These values are comparable to those reported previously, confirming the functional expression of ABCB1 in the Caco-2 cells [41]. The antiretrovirals atazanavir (50 µM), darunavir (100 µM), lopinavir (50 µM), rilpivirine (20 µM), ritonavir (50 µM), and saquinavir (20 µM), as well as the DAAs asunaprevir (50 µM), daclatasvir (20 µM), and grazoprevir (50 µM), all reduced the rP_app_ of [^3^H]-digoxin to values in the range of 1.11 to 1.91, making their effects comparable to that of the model inhibitor. Atazanavir (20 µM), darunavir (50 µM), etravirine (20 µM), lopinavir (5 µM), and ritonavir (20 µM) and DAA asunaprevir (20 µM), elbasvir (5 µM), grazoprevir (20 µM), and ledipasvir (50 µM) also significantly inhibited [^3^H]-digoxin, albeit to a lesser extent, giving rP_app_ values of 2.75 to 5.88. Etravirine, elbasvir, and ledipasvir exhibited low solubility, so higher concentrations that could potentially inhibit ABCB1 more strongly were not tested. No significant effect on [^3^H]-digoxin transport was observed for the antiretrovirals abacavir, dolutegravir, maraviroc, tenofovir DF, and zidovudine, or the DAAs sofosbuvir and velpatasvir. Whereas abacavir, maraviroc, and tenofovir DF were tested at the highest chosen concentration of 100 µM, dolutegravir and velpatasvir were only tested at concentrations of 10 and 5 µM, respectively, due to their limited solubility. These results are summarized in Table 1 (antiretrovirals) and Table 2 (DAAs). Papp values for apical (A) to basolateral (B) and B to A transports are summarized in Appendix A.

### 2.2. Effect of Antiretrovirals and DAA on ATP Content in hPCIS

Because the validity of accumulation studies in hPCIS strongly depends on hPCIS viability, we investigated the effect of 2.5 h treatments with [^3^H]-digoxin (0.3 µCi/mL; 15 nM) together with antiretrovirals, DAAs, or CP100356 on the ATP content of hPCIS prepared from intestine samples collected from four donors. There were no statistically significant differences between the ATP contents of the tested samples (Figure 1). The median ATP concentrations detected in hPCIS exposed to antiretrovirals ranged from 4.5 to 5.7 pmol × µg^−1^, while those in DAA-treated hPCIS were between 5.5 and 5.4 pmol × µg^−1^. For comparative purposes, the ATP contents of control hPCIS and hPCIS exposed to CP100356 (2 µM) were 6.1 and 6.6 pmol × µg^−1^, respectively. It thus appears that the model inhibitor and antivirals did not affect PCIS viability, even at the highest tested concentrations.

### 2.3. Effect of Antiretrovirals and DAA on [^3^H]-Digoxin Accumulation in hPCIS

To investigate the inhibitory effect of antivirals on ABCB1 in the intestine, we used hPCIS prepared from jejunal tissue obtained from five donors. The model ABCB1 inhibitor CP100356 (2 µM) increased the accumulation of [^3^H]-digoxin 12-fold. As shown in Figure 2A, the uptake of [^3^H]-digoxin increased when hPCIS were treated with atazanavir (50 µM; 9.2-fold), darunavir (100 µM, 4.0-fold), lopinavir (50 µM, 5.0-fold), ritonavir (20 and 50 µM; 4.5- and 5.0-fold, respectively), and saquinavir (20 µM, 4.0-fold). The observed increases were comparable to those observed for CP100256 (2 µM). In contrast to the results obtained in Caco-2 cells, atazanavir (20 µM), darunavir (50 µM), etravirine (20 µM), and rilpivirine (20 µM) did not inhibit [^3^H]-digoxin efflux from hPCIS. Abacavir (100 µM), dolutegravir (10 µM), lopinavir (20 µM), maraviroc (100 µM), saquinavir (5 µM), tenofovir DF (100 µM), and zidovudine (100 µM) also did not induce any detectable inhibition of [^3^H]-digoxin efflux, in accordance with the results obtained using Caco-2 cells.

In keeping with the observation in Caco-2 cells, the DAAs asunaprevir (at both tested concentrations of 20 and 50 µM) and daclatasvir (at 20 µM) increased the [^3^H]-digoxin uptake by factors of 9.3, 15.2, and 13.2, respectively, whereas sofosbuvir (100 µM) had no effect. However, in contrast to the results obtained in Caco-2 cells, grazoprevir increased [^3^H]-digoxin uptake only at the highest tested concentration (50 µM); the other compounds showing inhibitory activity in vitro, i.e., elbasvir (5 µM) and ledipasvir (20 and 50 µM), caused no inhibition of [^3^H]-digoxin efflux from hPCIS. Conversely, velpatasvir (5 µM) caused a significant (9.3-fold) increase in [^3^H]-digoxin accumulation in hPCIS.

## 3. Discussion

Combination antiretroviral therapy and DAA treatment regimens are a highly effective standard pharmacotherapy for HIV and HCV infections, respectively [11,42]. However, they frequently give rise to DDI with other antiretrovirals, DAA, or drugs used to treat comorbidities [15,43,44,45]. Antiretrovirals and DAA are known to interact with ABCB1 [16,17,18], but their ability to inhibit ABCB1 directly in the human intestine has not previously been studied in detail.

We have previously tested the inhibitory effect of antiretrovirals and DAA on the intestinal ABCB1-mediated transport of the fluorescent probe RHD123 [36] using a combination of bidirectional transport studies in Caco-2 cells and accumulation assays in PCIS [34]. The main advantages of RHD123 are its low cost, easy detection, and relatively low toxicity [36]. However, relying exclusively on RHD123 as an ABCB1 probe may prevent the detection of ABCB1 inhibitors that bind to other transporter-competent sites [20,21,22]. Digoxin, the probe used in this study, is frequently prescribed by clinicians despite its narrow therapeutic index and high frequency of DDI, including with antiviral drugs [11,15,42,46]. Importantly, it is considered to be a sensitive substrate for testing ABCB1 transport and the inhibition of ABCB1-mediated efflux in cell lines [39,47] because it undergoes minimal metabolism and exhibits low inhibitory potency towards clinically relevant intestinal transporters other than ABCB1 [38,48]. The use of digoxin instead of RHD123 in inhibition studies increases the number of identified inhibitors, probably because it has a larger binding region that partially overlaps with the R-site [20,21,49].

As stated previously, hPCIS prepared from human jejunum were used in this study. The jejunum has a very high absorptive capacity and a high expression of ABCB1 [50]. The ileum has similar characteristics [51,52], but tests on this segment were not performed because, at the time of writing the manuscript, it was impossible to collect healthy segments of the human ileum from the University Hospital in Hradec Kralove. Viability was assessed via an ATP content analysis in control and drug-exposed hPCIS, as previously recommended [33,34,35].

Of the antiretrovirals tested, atazanavir, darunavir, etravirine, lopinavir, rilpivirine, ritonavir, and saquinavir inhibited ABCB1-mediated digoxin transport in Caco-2 cells and/or hPCIS. The inhibitory effects of atazanavir, darunavir, ritonavir, and saquinavir are consistent with results from previous studies using other in vitro experimental models [37,53,54] and with reported increases in the AUC of digoxin and dabigatran etexilate in vivo [11]. The inhibitory activity of atazanavir, ritonavir, and saquinavir was also previously observed in Caco-2 cells and hPCIS when using RHD123 as the probe [34]. Furthermore, in accordance with results obtained using non-intestinal experimental models [54,55], we found that lopinavir is a potent inhibitor of the ABCB1-mediated transport of RHD123 [34] and digoxin in Caco-2 cells and hPCIS (Table 1 and Figure 2A). Similarly, a docking analysis using a mice model of ABCB1a (PDB code: 4M1M, Appendix A) showed large binding contact in an ABCB1a cavity (Appendix A) and binding free energy (Appendix A). Surprisingly, lopinavir does not alter the pharmacokinetics of ABCB1 substrates in vivo [11]. This is probably because prolonged exposure to lopinavir increases ABCB1 expression, which compensates for its inhibitory activity. As a result, overall reductions in ABCB1 activity following lopinavir treatment are only observed after acute exposure [56]. Rilpivirine was also previously suggested to inhibit ABCB1 in vitro [11,57]. However, rilpivirine at a dose of 25 mg/day does not significantly affect the pharmacokinetics of digoxin or tenofovir DF in vivo [57], suggesting that the inhibition of ABCB1 in human tissues by itself is insufficient to change the pharmacokinetics of ABCB1 substrates. In keeping with this hypothesis, rilpivirine did not affect digoxin efflux in hPCIS (Figure 2A). Additionally, our results indicate that etravirine inhibits ABCB1 in Caco-2 monolayers, contradicting results obtained previously using cell line models with calcein, pheophorbide A, and RHD123 as probes [37,58]. Presumably, the inhibitory effects of etravirine towards ABCB1 probe substrates differ because of the presence of multiple substrate binding sites in ABCB1 [20,21,49]. However, etravirine had no significant inhibitory effect in experiments using hPCIS. We therefore hypothesize that it is only a weak ABCB1 inhibitor, in accordance with the information provided in the summary of product characteristics for the drug Intelence, in which etravirine is the active ingredient [59], and a docking analysis (Appendix A).

Abacavir, dolutegravir, tenofovir DF, and zidovudine have been identified as likely ABCB1 substrates [60,61,62,63]. This identification is supported by their relatively high free energies of binding to the transporter (see Appendix A). In keeping with previous studies [11,37], we observed no ABCB1 inhibitory activity for abacavir, tenofovir DF, or zidovudine. Therefore, these antiretrovirals are unlikely to compete with digoxin for binding to its ABCB1 binding pocket or to affect ABCB1 function by binding to the access tunnel [64]. The docking analysis also showed a narrower contact of abacavir with the binding cavity when compared with lopinavir (Appendix A). We also did not observe any inhibitory activity of dolutegravir (Table 1, Figure 2A), contradicting a previous suggestion that it might be a weak inhibitor of digoxin transport in MDCK-ABCB1 cells based on apparent activity at a concentration of 100 µM [60]. However, the reported solubility of this drug in dimethyl sulfoxide (DMSO) is poor; its maximum dissolved concentration is claimed to be in the range of 5 to 10 mM. Therefore, we prepared the stock solution at a concentration of 10 mM. To avoid exceeding the maximum DMSO concentration of 0.1% in the test solution, we could only test dolutegravir at concentrations of up to 10 µM. Because the concentration of dolutegravir could potentially exceed 100 µM in the intestine, an inhibitory effect on intestinal ABCB1 in patients cannot be ruled out based on our results.

Of the tested DAA, asunaprevir, daclatasvir, and grazoprevir inhibited ABCB-mediated digoxin transport in Caco-2 cells and hPCIS. Asunaprevir and daclatasvir have been previously suggested to inhibit ABCB1 in vitro [34,65], and both drugs increase digoxin bioavailability in humans [15,66]. Velpatasvir inhibited ABCB1 only in hPCIS. This finding is consistent with observations in healthy volunteers, in whom velpatasvir increased digoxin exposure by 34% without changing its t_1/2_ [67]. Ledipasvir inhibited RHD123 transport in Caco-2 cells [34] and is suggested to modestly increase the AUC of digoxin in humans [42]. Our data, obtained using hPCIS, thus support the conclusion that asunaprevir, daclatasvir, and ledipasvir increase the digoxin AUC by inhibiting intestinal ABCB1. Elbasvir also inhibited ABCB1 in vitro [42], but its effect on intestinal ABCB1 inhibition appears to be minimal in humans (11% increase in plasma AUC) [42], which is consistent with its lack of effect in our hPCIS experiments (Figure 2B). Although grazoprevir is a substrate for ABCB1 [42], it was reported not to inhibit ABCB1 in vitro [68], contradicting the results obtained with both of our experimental models. Unfortunately, the studies in which this inhibitory activity was observed are not publicly accessible, making it impossible to know what concentrations were tested or which experimental models were used. The concentrations of grazoprevir tested in our assays could plausibly occur in the small intestine during therapy, so their effects on the intestinal absorption of ABCB1 substrates in humans warrant evaluation. Sofosbuvir is also a substrate of ABCB1 that does not appear to inhibit ABCB1 [42]. Our experiments using digoxin (Table 2, Figure 2B) and RHD123 [34] as probes confirmed this compound’s non-inhibition of ABCB1. Since the calculated free energy of binding of sofosbuvir to ABCB1 is similar to that of digoxin (Appendix A), it can be speculated that its mode of binding to ABCB1 differs from that of typical inhibitors [64].

Some antivirals exhibited different inhibitory potencies towards ABCB1-mediated digoxin efflux in Caco-2 cells and hPCIS (Table 1 and Table 2 and Figure 2). Etravirine, rilpivirine, elbasvir, and ledipasvir inhibited digoxin transport in Caco-2 cells but had no effect on digoxin accumulation in hPCIS. Because Caco-2 cells and human jejunum express comparable levels of ABCB1 [50,69], we hypothesize that this outcome is due to the previously suggested greater sensitivity of bidirectional transport studies [70], which results from the reduced binding of test compounds to cell membranes [70] and the narrower tight junctions in the Caco-2 system [71]. Furthermore, differences in the expression of metabolic enzymes and uptake transporters between the Caco-2 cell line and hPCIS could also explain these discrepancies. The hPCIS, compared to the Caco-2 cell line, express a cytochrome P450 3A4 (CYP3A4) drug-metabolizing enzyme [50]. Some of the tested antivirals can be extensively metabolized by CYP3A4 [72], resulting in less pronounced ABCB1 inhibition. On the other hand, metabolites that are produced by enzymatic conversion can also inhibit ABCB1 [73]. On the contrary, the Caco-2 cell line expresses higher levels of some uptake transporters, which could lead to the increased intracellular concentration of the antivirals, leading to the more significant ABCB1 inhibition [50,74]. The hPCIS-based model was previously also found to be less sensitive than the Caco-2 system when measuring the inhibitory potency of maraviroc and ledipasvir [34]. On the other hand, velpatasvir had no effect on digoxin transport in Caco-2 cells, but inhibited ABCB1 in hPCIS. However, it should be noted that, due to its low solubility, velpatasvir was only tested at a concentration of 5 µM, even though its suggested IC_50_ for ABCB1 is approximately 20.6 µM [67]. We hypothesize that its lack of effect in Caco-2 cells was due to the extensive binding of velpatasvir to bovine serum albumin in the acceptor compartment, which would reduce the amount of free velpatasvir present in bidirectional experiments. Alternatively, velpatasvir metabolites produced in hPCIS by CYP3A4 [42] could be responsible for the ABCB1 inhibition seen in that model system.

The use of digoxin as the probe in these bidirectional transport studies led to the identification of etravirine and rilpivirine as ABCB1 inhibitors (Table 3 and Table 4), neither of which were identified as inhibitors when using RHD123 as the probe [34]. This is consistent with evidence that ABCB1-mediated digoxin transport appears to be affected by a wide spectrum of clinically relevant compounds [20,21]. However, maraviroc significantly affected the ABCB1-mediated transport of RHD123 [34], but not digoxin, in Caco-2 cells (Table 3). The docking analysis showed that maraviroc binds to ABCB1 with a free energy of binding of −8.85 kcal/mol, which is similar to that for digoxin (−8.55 kcal/mol) and greater than that for RHD123 (−6.74 kcal/mol; see Appendix A). The absence of DDI between maraviroc and digoxin is also supported by clinical findings showing that total exposure to digoxin was unaffected by the presence of maraviroc in healthy volunteers [75].

## 4. Materials and Methods

### 4.1. Reagents and Chemicals

[^3^H]-digoxin was purchased from Moravek Biochemicals (Brea, CA, USA). Abacavir, atazanavir, etravirine, lopinavir, maraviroc, rilpivirine, ritonavir, saquinavir, tenofovir DF, and zidovudine were obtained from the NIH AIDS Reagent Program. Asunaprevir, daclatasvir, darunavir, dolutegravir, elbasvir, grazoprevir, ledipasvir, sofosbuvir, and velpatasvir were acquired from MedChemExpress LLC (Middlesex County, NJ, USA). The model ABCB1 inhibitor CP100356 [76], the ATP bioluminescence assay kit, DMSO, Dulbecco’s Modified Eagle Medium, ethanol (EtOH), fetal bovine serum, nonessential amino acid solution, and penicillin–streptomycin solutions were purchased from Sigma-Aldrich (St. Louis, MO, USA). Hanks’ balanced salt solution, William’s medium E containing L-glutamine (WME), and the bicinchoninic acid protein assay kit were obtained from Thermo Fisher Scientific (Waltham, MA, USA). Krebs–Henseleit buffer was prepared as described by de Graaf et al. [31]. All other reagents were of analytical grade.

### 4.2. Stock Solutions and Test Solutions

CP100356 and all antivirals were dissolved in DMSO, while digoxin was dissolved in 99.9% EtOH. The stock solutions were stored at −20 °C before use. The final concentration of DMSO was 0.1% in all experiments. The final concentrations of EtOH in hPCIS and Caco-2 cells were 0.03% and 0.012%, respectively. The highest tested concentrations of the antivirals were determined by their solubility in the incubation medium; if solubility was not limiting, the maximum tested concentration was the lowest concentration at which the rPapp became comparable to that achieved with CP100356.

### 4.3. Cell Culture and Growth Condition

The Caco-2 colorectal adenocarcinoma cell line (ATCC HTB-37) was purchased from the American Type Culture Collection and cultured in high-glucose Dulbecco’s Modified Eagle Medium, with L-glutamine supplemented with 10% fetal bovine serum and a 1% nonessential amino acid solution. Cells were routinely cultured in an antibiotic-free medium and incubated in a humidified incubator under a 5% CO_2_ atmosphere at 37 °C. Cells from passages 10 to 40 were used in all in vitro experiments.

### 4.4. Human Tissue Samples

Intestinal samples (jejunum) were collected from five donors (Table 5) while they were undergoing the Whipple procedure (pancreaticoduodenectomy) at the University Hospital in Hradec Kralove, Czech Republic. Sample collection was performed with written informed patient consent and the approval of the local research ethics committee (approval no. 201511 S26P and 202103 I67P) [34,77].

### 4.5. In Vitro Bidirectional Permeability Experiments

Transport experiments were performed using microporous polycarbonate membrane filters (0.4 µm pore size, 12 mm diameter; Transwell^®^ 3401; Costar, Corning, NY, USA), as previously described [34]. Caco-2 cells were seeded at a density of 3 × 10^5^ cells per insert and cultured for 21 days in a standard cultivation medium containing 1% penicillin–streptomycin. The medium was changed every other day, during which the transepithelial electrical resistance (TEER) across the cell monolayers was measured using a Millicell-ERS instrument (Millipore Corporation, Bedford, MA, USA) [34]. The TEER values before the start of the experiment ranged from 1100 to 1900 Ωcm^2^, which is consistent with previous reports [78,79]. For the bidirectional permeability assay, a Hanks’ balanced salt solution buffer was used. The pH in the A compartment was adjusted to 6.5 using a methanesulfonic acid solution, while that in the B compartment was adjusted to 7.4 using a HEPES solution [34,80]. The volumes used were 0.5 mL and 1.5 mL in compartments A and B, respectively. To improve the reproducibility of the results, the receiver compartment always contained 1% bovine serum albumin, as previously recommended [80]. All wells were preincubated for 30 min with the appropriate transport buffer, containing CP100356 or one of the tested antiviral drugs [34,80]. The assay was started by placing fresh buffer containing CP100356 or an antiviral drug in the donor compartment (compartment A for A to B transport and compartment B for B to A transport) together with [^3^H]-digoxin at an activity level of 0.12 µCi/mL, corresponding to a low non-saturating concentration of 6 nM [81]. Samples (200 µL) were collected after 1 and 2 h from the receiver compartment; after the first collection, fresh receiver solution was added to the receiver compartment to maintain the original volume [80]. The concentration of [^3^H]-digoxin was quantified using a Tri-Carb 2900TR liquid scintillation analyzer (Packard Bioscience, Meriden, CT, USA). Its concentration in the samples collected during Caco-2 experiments was measured after adding 1 mL of Ultima Gold^TM^ Cocktail. A to B and B to A transport were evaluated in terms of an apparent permeability coefficient (P_app_), calculated using the equation in [80,81].
P_app_ = (dC/dt) × V_r_/(A × C_0_)(1)
where dC/dt is the change in concentration over time measured during the linear phase of transport over 1 h, V_r_ is the volume of the receiver well in milliliters, A is the area of the membrane in square centimeters, and C_0_ is the initial concentration in the donor compartment. The efflux ratio (rP_app_) was then calculated using the equation [80,82]:rP_app_ = (P_app_ B to A)/(P_app_ A to B)(2)

### 4.6. Analysis of the ATP Content in hPCIS

To evaluate whether antivirals or [^3^H]-digoxin at an activity of 0.3 µCi/mL (15 nM) had any impact on the viability of the hPCIS, the intracellular ATP content, which is a verified marker of the preservation of vital cell processes [35], was measured using the CLS II ATP bioluminescence assay kit (Roche, Mannheim, Germany), as previously described [34,35,83]. Measurements were performed using fresh hPCIS after 2.5 h incubation with [^3^H]-digoxin (0.3 µCi/mL; 15 nM) and/or the antivirals. The ATP contents before and after incubation were then compared.

### 4.7. Ex Vivo Accumulation Experiments in hPCIS Prepared from the Jejunum

Ex vivo accumulation assays were performed as previously described [32]. Directly after surgery, the resected part of the human jejunum was placed in cold (4 °C) Krebs–Henseleit buffer oxygenated with carbogen gas [32,34,35]. The muscle layer was removed, and the remaining mucosa was cut into fragments measuring approximately 5 by 20 mm, which were then embedded in a 3% agarose solution (3% (wt/vol) in 0.9% NaCl, 37 °C). PCIS of approximately 300 µm thickness were cut using a Krumdieck tissue slicer (Alabama R&D, Munford, AL, USA). Slices were pre-incubated for 30 min in the presence of CP100356 or an antiviral drug in WME [34,35] and then transferred to a WME incubation medium containing [^3^H]-digoxin (0.3 µCi/mL; 15 nM) and the compound being tested. Both incubation steps were conducted in a humidified atmosphere of 80% O_2_ and 5% CO_2_ at 37 °C [32,34,35]. The accumulation of [^3^H]-digoxin was stopped after 2 h of incubation by washing the slices twice in the Krebs–Henseleit cold buffer (4 °C). Slices were transferred to 2 mL microvials containing 600 µL of acetonitrile solution (acetonitrile/water ratio, 2:1) and approximately 300 mg of glass minibeads (diameter, 1.25 to 1.65 mm; Carl Roth, Karlsruhe, Germany), and homogenized with a FastPrep24 5G minibead beater (MP Biomedicals, Santa Ana, CA, USA; 6.0 m/s, twice for 45 s each). The samples were then centrifuged (10 min; 7800 g). Concentrations of [^3^H]-digoxin in the supernatant samples (300 µL) were assessed after mixing them with 1.5 mL of Ultima Gold™ via scintillation counting (Tri-Carb 2900TR liquid scintillation analyzer, Packard Bioscience). The pellets obtained during centrifugation were dried overnight at 37 °C and then solubilized in 200 µL of 5 M NaOH for 24 h. Milli-Q water was then added to the samples to achieve a NaOH concentration of 1 M. The protein content was measured using a bicinchoninic acid protein kit (Thermo Fisher Scientific, Waltham, MA, USA). The measured [^3^H]-digoxin concentrations were normalized against the protein content.

### 4.8. Statistical Evaluation

Statistical analysis of bidirectional transport across Caco-2 cells was performed using an ordinary one-way ANOVA with Dunnett’s post hoc multiple comparisons test. The statistical significance of differences in the measured ATP contents of hPCIS after different treatments was assessed using the non-parametric Kruskal–Wallis test followed by Dunn’s test. The effect of antivirals on [^3^H]-digoxin accumulation in hPCIS was evaluated using the non-parametric paired Friedman test followed by Dunn’s test. Values differing from controls at the *p* < 0.05, *p* < 0.01, and *p* < 0.001 levels are indicated with the labels *, **, and ***, respectively.

## 5. Conclusions

We have shown that asunaprevir, atazanavir, daclatasvir, darunavir, grazoprevir, lopinavir, ritonavir, and saquinavir inhibit ABCB1 in both Caco-2 monolayers and hPCIS. Therefore, we conclude that these drugs have a high potential to cause DDIs on intestinal ABCB1. Our hPCIS data also suggest that velpatasvir may inhibit intestinal ABCB1. There is supporting clinical evidence that most of these antivirals increase the AUC of ABCB1 substrates. Our findings suggest that the inhibition of intestinal ABCB1 contributes to this increase in AUC and should, therefore, be taken into account when establishing new antiviral combination regimens or when considering polypharmacy in HIV- and/or HCV-positive patients, especially in cases involving drugs whose absorption is significantly reduced by ABCB1, such as dabigatran etexilate and digoxin [11,15]. On the other hand, abacavir, dolutegravir, maraviroc, sofosbuvir, tenofovir DF, and zidovudine exhibit no apparent inhibitory activity towards ABCB1. Comparing the two experimental models used in this work, we conclude that the bidirectional transport-based assay using Caco-2 cells is more sensitive and better able to reveal ABCB1 inhibition, but that the results of hPCIS experiments agree more closely with published in vivo findings. Additionally, more inhibitors are identified when using digoxin as the ABCB1 probe substrate than when using RHD123. However, both probes have limitations, so inhibitory potency should be tested using at least two ABCB1 probes.

## Figures and Tables

**Figure 1 pharmaceuticals-15-00242-f001:**
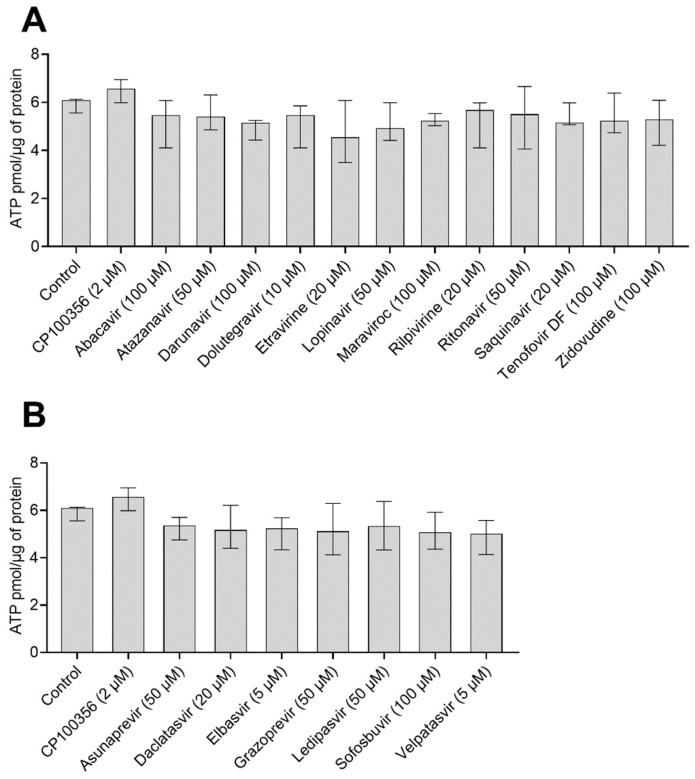
ATP contents of hPCIS (*n* = 4) after 2.5 h of incubation with [^3^H]-digoxin in the presence of the studied (**A**) antivirals and (**B**) DAA at their highest tested concentrations. Data are presented as medians with interquartile ranges. Statistical significance was assessed using the nonparametric Kruskal–Wallis test followed by Dunn’s test. No statistically significant differences were found.

**Figure 2 pharmaceuticals-15-00242-f002:**
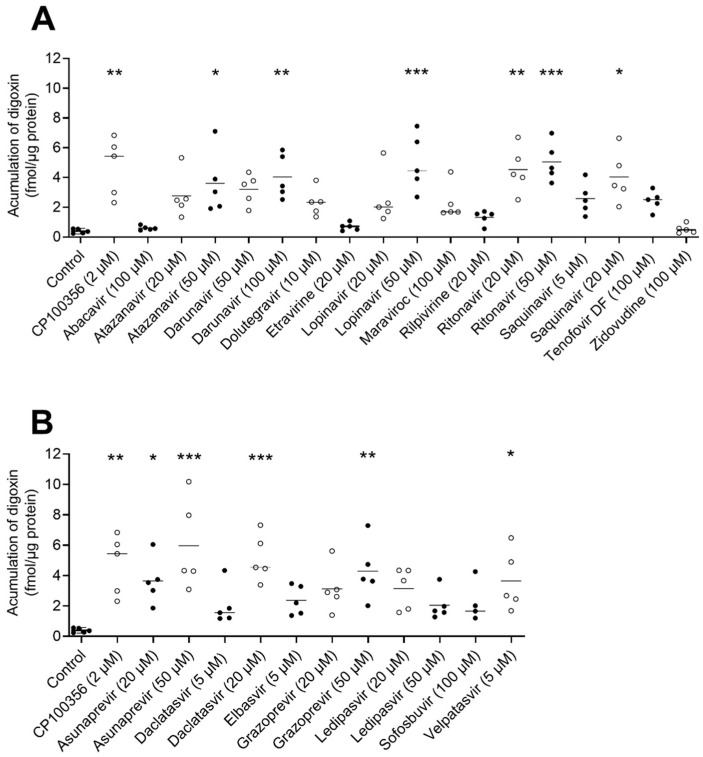
Effects of selected (**A**) antiretrovirals and (**B**) DAA on [^3^H]-digoxin accumulation in hPCIS. Data are presented as medians (*n* = 5). Statistical analysis was performed using the nonparametric paired Friedman test followed by Dunn’s test: *p* < 0.05 (*); *p* < 0.01 (**); *p* < 0.001 (***).

**Table 1 pharmaceuticals-15-00242-t001:** Effects of the antiretrovirals and the model inhibitor CP100356 on ABCB1-controlled [^3^H]-digoxin transport across Caco-2 monolayers.

Compound	Concentration	[^3^H]-Digoxin rP_app_ ^1^
Control	6 nM	9.53 ± 2.22
+CP100356	2 µM	1.49 ± 0.11 ***
+Abacavir	100 µM	10.39 ± 2.35
+Atazanavir ^2^	20 µM	5.57 ± 0.81 *
	50 µM	1.15 ± 0.22 ***; #
+Darunavir	20 µM	6.19 ± 1.83
	50 µM	3.28 ± 0.39 ***
	100 µM	1.74 ± 0.26 ***; #
+Dolutegravir ^3^	10 µM	11.91 ± 2.05
+Etravirine ^3^	20 µM	3.23 ± 0.41 ***
+Lopinavir ^2^	5 µM	5.24 ± 1.69 *
	50 µM	1.91 ± 0.23 ***; #
+Maraviroc	20 µM	11.25 ± 0.11
	100 µM	8.80± 1.26
+Rilpivirine ^2^	20 µM	1.52 ± 0.53 ***
+Ritonavir ^2^	20 µM	2.75 ± 0.97 ***
	50 µM	1.11 ± 0.10 ***
+Saquinavir ^2^	5 µM	8.50 ± 3.30
	20 µM	1.36 ± 0.20 ***; #
+Tenofovir DF	100 µM	11.76 ± 0.07
+Zidovudine	100 µM	13.42 ± 0.28

^1^ rP_app_, efflux ratio. Statistical analysis was performed using an ordinary one-way ANOVA with Dunnett’s post hoc multiple comparisons test. Values differing significantly from the control are indicated by the labels * (*p* < 0.05) or *** (*p* < 0.001). Values differing significantly from those obtained with the same compound at a lower concentration are indicated by the labels # (*p* < 0.05). ^2^ Higher concentrations were not tested because an rPapp of approximately 1 was reached. ^3^ Higher concentrations were not tested due to limited solubility.

**Table 2 pharmaceuticals-15-00242-t002:** Effects of the tested DAAs and the model inhibitor CP100356 on ABCB1-controlled [^3^H]-digoxin transport across Caco-2 monolayers.

Compound	Concentration	[^3^H]-Digoxin rP_app_ ^1^
Control	6 nM	9.53 ± 2.22
+CP100356	2 µM	1.49 ± 0.11 ***
+Asunaprevir ^2^	20 µM	3.07 ± 0.52 ***
	50 µM	1.27 ± 0.18 ***
+Daclatasvir ^2^	5 µM	9.75 ± 0.43
	20 µM	1.22 ± 0.33 ***; ###
+Elbasvir ^3^	5 µM	5.88 ± 1.01 *
+Grazoprevir ^2^	20 µM	3.79 ± 0.27 ***
	50 µM	1.21 ± 0.15 ***
+Ledipasvir ^3^	20 µM	9.39 ± 1.76
	50 µM	3.96 ± 0.90 **; #
+Sofosbuvir	100 µM	6.09 ± 0.18
+Velpatasvir ^3^	5 µM	7.38 ± 1.81

^1^ rP_app_, efflux ratio. Statistical analysis was performed using an ordinary one-way ANOVA with Dunnett’s post hoc multiple comparisons test. Values differing significantly from the control are indicated by the labels * (*p* <0.05), ** (*p* < 0.01), or *** (*p* < 0.001). Values differing significantly from those obtained with the same compound at a lower concentration are indicated by the labels # (*p* < 0.05) or ### (*p* < 0.001). ^2^ Higher concentrations were not tested because an rPapp of approximately 1 was reached. ^3^ Higher concentrations were not tested due to limited solubility.

**Table 3 pharmaceuticals-15-00242-t003:** Inhibition of bidirectional transport of the probes digoxin and RHD123 [34] across monolayers of Caco-2 cells in the presence of various drugs.

Compound	Concentration	Digoxin Inhibition	RHD123 Inhibition #
CP100356	2 µM	YES	YES
Abacavir	100 µM	NO	NO
Atazanavir	50 µM	YES	YES
Daclatasvir	20 µM	YES	YES
Etravirine	20 µM	YES	NO
Ledipasvir	50 µM	YES	YES
Lopinavir	5 µM	YES	YES
Maraviroc	100 µM	NO	YES
Rilpivirine	20 µM	YES	NO
Ritonavir	50 µM	YES	YES
Saquinavir	20 µM	YES	YES
Sofosbuvir	100 µM	NO	NO
Tenofovir DF	100 µM	NO	NO

RHD123, rhodamine123; # results are taken from [34].

**Table 4 pharmaceuticals-15-00242-t004:** Inhibition of digoxin and RHD123 transport in human PCIS by various drugs.

Compound	Concentration	Digoxin Inhibition	RHD123 Inhibition #
CP100356	2 µM	YES	YES
Atazanavir	50 µM	YES	YES
Daclatasvir	20 µM	YES	NO *
Ledipasvir	50 µM	NO	NO *
Lopinavir	50 µM	YES	YES
Maraviroc	100 µM	NO	NO *
Ritonavir	100 µM	YES	YES
Saquinavir	20 µM	YES	YES

RHD123, rhodamine123; * increased uptake of RHD123 was observed in some samples, but the median did not differ significantly from that of the control; # results are taken from [34].

**Table 5 pharmaceuticals-15-00242-t005:** Characteristics of intestinal donors.

Patient No.	Gender	Age (Year)	Medication(s)
1	F	62	candesartan, levothyroxine
2	F	71	diosmin, flavonoids
3	F	73	apixaban, atorvastatin, betaxolol, omeprazole, pancreatin, ramipril, rilmenidine
4	F	49	dosulepin, lactulose, pancreatin, pantoprazole, pregabalin, thiamine, trazodone,
5	M	74	acetylsalicylic acid, amlodipine, budesonide, flavonoids, ipratropium bromide, levothyroxine, metformin, omeprazole, tamsulosin, telmisartan

## Data Availability

The data presented in this study are available in this article or Appendix A.

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
