# Peer review of "Evaluation of the Potency of Anti-HIV and Anti-HCV Drugs to Inhibit P-Glycoprotein Mediated Efflux of Digoxin in Caco-2 Cell Line and Human Precision-Cut Intestinal Slices"

_pharmaceuticals, 2022, doi:10.3390/ph15020242_

Round 1

Reviewer 1 Report

  1. Whether the drugs at high conc. influence the viability of Caco-2 cells?
  2. The concentrations used for some drugs are pretty high, not relevant to the clinical dose. Thus the finding looks weak on this means.
  3. Will there any DDI between the drugs and other transporters (PEPT2)or CYP450. 
  4. Both Papp values for A to B and B to A should be given in Tables, instead of just efflux ratio was given.
  5. Why PH for the buffer was adjusted to 6.5. 
  6. "those in in DAA-treated" in line 130 should revised.

Author Response

 Answers to the reviewer 1

Dear reviewer,

Thank you for your valuable comments and questions. Our answers can be found below (also attached as a file). We hope you find our answers satisfactory.  

Authors

Question 1: Whether the drugs at high conc. influence the viability of Caco-2 cells?

Answer: The transepithelial electrical resistance (TEER) was measured before and after the Caco-2 experiment to determine the intactness of the monolayer as previously recommended (Hubatsch et al. 2007). We did not observe a decrease in TEER values confirming that tested concentrations of antiretrovirals did not disrupt the integrity of monolayers. To further support the experimental setup, we here refer to publications in which comparable concentrations of antiviral agents were used (Kis et al., 2013; Holmstock et al., 2012; Profit et al., 1999; Storch et al., 2007; Tong et al., 2007).

Kis, Olena, Jason A. Zastre, Md. Tozammel Hoque, Sharon L. Walmsley, and Reina Bendayan. “Role of Drug Efflux and Uptake Transporters in Atazanavir Intestinal Permeability and Drug-Drug Interactions.” Pharmaceutical Research 30, no. 4 (April 7, 2013): 1050–64. https://doi.org/10.1007/s11095-012-0942-y.

Holmstock, Nico, Pieter Annaert, and Patrick Augustijns. “Boosting of HIV Protease Inhibitors by Ritonavir in the Intestine: The Relative Role of Cytochrome P450 and P-Glycoprotein Inhibition Based on Caco-2 Monolayers versus In Situ Intestinal Perfusion in Mice.” Drug Metabolism and Disposition 40, no. 8 (August 2012): 1473–77. https://doi.org/10.1124/dmd.112.044677.

Hubatsch, I.; Ragnarsson, E.G.; Artursson, P. Determination of drug permeability and prediction of drug absorption in Caco-2 monolayers. Nat Protoc 2007, 2, 2111-2119, doi:10.1038/nprot.2007.303

Profit, Louise, Victoria A. Eagling, and David J. Back. “Modulation of P-Glycoprotein Function in Human Lymphocytes and Caco-2 Cell Monolayers by HIV-1 Protease Inhibitors.” AIDS 13, no. 13 (September 1999): 1623–27. https://doi.org/10.1097/00002030-199909100-00004.

Storch, Caroline Henrike, Dirk Theile, Heike Lindenmaier, Walter Emil Haefeli, and Johanna Weiss. “Comparison of the Inhibitory Activity of Anti-HIV Drugs on P-Glycoprotein.” Biochemical Pharmacology 73, no. 10 (May 2007): 1573–81. https://doi.org/10.1016/j.bcp.2007.01.027.

Tong, Leah, Truc K. Phan, Kelly L. Robinson, Darius Babusis, Robert Strab, Siddhartha Bhoopathy, Ismael J. Hidalgo, Gerald R. Rhodes, and Adrian S. Ray. "Effects of Human Immunodeficiency Virus Protease Inhibitors on the Intestinal Absorption of Tenofovir Disoproxil Fumarate in Vitro." Antimicrobial Agents and Chemotherapy 51, no. 10 (2007): 3498–3504. https://doi.org/10.1128/AAC.00671-07.

Question 2:  The concentrations used for some drugs are pretty high, not relevant to the clinical dose. Thus, the finding looks weak on this means.

Answer: European Medicine Agency (2012) and Food and Drug Administration (2020) recommend the volume of 250 ml to predict drug concentrations achieved in the intestine. When considering the prescribed doses, achieved intestinal concentrations of antiretrovirals significantly exceed the tested concentrations in our study (see Table 1 in Response to Reviewers). Therefore, even higher concentrations should be tested than those used in our study. However, it was not possible to do so due to limited solubility of antivirals and cytotoxicity of the solvent, DMSO, in concentrations (v/v) higher than 0.1%.

We discussed it specifically for dolutegravir, because the maximal concentration tested was 10 µM due to its low solubility in DMSO. However, dolutegravir tested at a concentration of 100 µM had been previously reported to be a weak inhibitor of ABCB1. Therefore, we discussed this issue in the manuscript as follows: ”However, the reported solubility of this drug [dolutegravir] in dimethylsulfoxide (DMSO) is poor; its maximum dissolved concentration is claimed to be in the range of 5 to 10 mM. We, therefore, prepared the stock solution at a concentration of 10 mM. To avoid exceeding the maximum DMSO concentration of 0.1% in the test solution, we could only test dolutegravir at concentrations of up to 10 µM. Because the concentration of dolutegravir could potentially exceed 100 µM in the intestine, an inhibitory effect on intestinal ABCB1 in patients cannot be ruled out based on our results.”

Just for the purpose of the reviewing process, we added Table 1 and Table 2 listing antivirals with tested concentrations and theoretical luminal concentrations.

EMA. “Guideline on the Investigation of Drug Interactions - Revision 1, CPMP/EWP/560/95/Rev.1 Corr.2,” 2012. www.ema.europa.eu/contact.

FDA. "In Vitro Drug Interaction Studies-Cytochrome P450 Enzyme-and Transporter-Mediated Drug Interactions Guidance for Industry," 2020. https://www.fda.gov/Drugs/GuidanceComplianceRegulatoryInformation/Guidances/default.htm.

Table 1.: Theoretical luminal concentrations achievable after single-dose administration – Anti-retrovirals.

Compound

Concentration used in our study (µM)

EMA approved dose (mg)

Mol. Weight (g/mol)

Teoretical luminal concentration (µM)

Abacavir

100 

300

286.33

4190.94

Atazanavir

20

300

704.86

1702.48

50

Darunavir

20

800

547.67

5842.97

50

100

Dolutegravir

10

50

419.39

476.89

Etravirine

20

200

435.29

1837.88

Lopinavir

200

628.814

1272.24

50 

Maraviroc

20

300

513.67

2336.15

100

Rilpivirine

20

25

366.42

272.91

Ritonavir

20

100

720.95

554.83

50

Saquinavir

5

500

670.84

2981.33

20

Tenofovir DF

100

245

635.50

1542.09

Zidovudine

100

250

267.24

3741.93

Table 2.: Theoretical luminal concentrations achievable after single-dose administration – Direct Acting Antivirals.

Compound

Concentration used in our study (µM)

EMA approved dose (mg)

Mol. Weight (g/mol)

Teoretical luminal concentration (µM)

Asunaprevir

20

100

748.29

534.55

50

Daclatasvir

5

60

738.89

324.81

20

Elbasvir

5

50

882.04

226.75

Grazoprevir

20

100

766.91

521.57

50

Ledipasvir

20

90

889.02

404.94

50

Sofosbuvir

100

400

529.45

3021.99

Velpatasvir

5

100

883.02

452.99

Question 3: Will there any DDI between the drugs and other transporters (PEPT2) or CYP450.

Answer: We aimed to demonstrate the inhibitory effect on ABCB1-mediated intestinal efflux of digoxin. Digoxin is a model substrate of ABCB1, recommended for testing drug-drug interactions by the International Transporter Consortium. In the intestine, digoxin is specifically transported by ABCB1. Furthermore, digoxin is not metabolized by CYP450 family enzymes in humans (Lacarelle et al., 1991, Taub et al., 2009). In conclusion, using our experimental setup, we detected exclusively antiviral-induced ABCB1 inhibition. As we observed differences in ABCB1 inhibition in Caco-2 cells and hPCIS, we discussed the metabolism of tested antiretrovirals in hPCIS as a potential reason of this phenomenon.

Antivirals included in our study likely interact with biotransformation enzymes and/or other transporters (Kis et al., 2010) located in enterocytes, but to reveal these DDIs, different substrates than digoxin would have to be used and experiments should be performed in hPCIS that are the mini-model of the intestine equipped with enzymes and transporters (de Graaf et al., 2010). However, this type of study was beyond the scope of this manuscript.

Graaf, Inge A M de, Peter Olinga, Marina H de Jager, Marjolijn T. Merema, Ruben de Kanter, Esther G van de Kerkhof, and Geny M M Groothuis. “Preparation and Incubation of Precision-Cut Liver and Intestinal Slices for Application in Drug Metabolism and Toxicity Studies.” Nature Protocols 5, no. 9 (September 19, 2010): 1540–51. https://doi.org/10.1038/nprot.2010.111.

Lacarelle, B., R. Rahmani, G. Sousa, A. Durand, M. Placidi, and JP Cano. “Metabolism of Digoxin, Digoxigenin Digitoxosides and Digoxigenin in Human Hepatocytes and Liver Microsomes.” Fundamental & Clinical Pharmacology 5, no. 7 (October 1991): 567–82. https://doi.org/10.1111/j.1472-8206.1991.tb00746.x.

Kis, Olena, Kevin Robillard, Gary N.Y. Chan, and Reina Bendayan. "The Complexities of Antiretroviral Drug–Drug Interactions: Role of ABC and SLC Transporters." Trends in Pharmacological Sciences 31, no. 1 (January 2010): 22–35. https://doi.org/10.1016/j.tips.2009.10.001.

Taub, Mitchell E., Kirsten Mease, Rucha S. Sane, Cory A. Watson, Liangfu Chen, Harma Ellens, Brad Hirakawa, Eric L. Reyner, Marton Jani, and Caroline A. Lee. "Digoxin Is Not a Substrate for Organic Anion-Transporting Polypeptide Transporters OATP1A2, OATP1B1, OATP1B3, and OATP2B1 but Is a Substrate for a Sodium-Dependent Transporter Expressed in HEK293 Cells." Drug Metabolism and Disposition 39, no. 11 (November 2011): 2093–2102. https://doi.org/10.1124/dmd.111.040816.

Question 4: Both Papp values for A to B and B to A should be given in Tables, instead of just efflux ratio was given.

 Answer: We have prepared a table summarizing the Papp values for A to B and B to A. We added this table to the supplementary materials (Tables S1 and S2) and references into the manuscript at line 102.  

Question 5:  Why PH for the buffer was adjusted to 6.5.

Answer: This was recommended by Hubatsch et al. in Nature protocols (2007). We referred to this paper in Methods paragraph 4.5. pH 6.5 should mimic the acidic microclimate of the small intestine. We already used this approach in our previous article, where rhodamine123 was used as a probe (Martinec et al., 2019). Besides, it is a recommended procedure, using the same experimental conditions allows us to compare the inhibition effects of antivirals on the efflux of two ABCB1 probes.

Hubatsch, Ina, Eva G E Ragnarsson, and Per Artursson. "Determination of Drug Permeability and Prediction of Drug Absorption in Caco-2 Monolayers." Nature Protocols 2, no. 9 (2007): 2111–19. https://doi.org/10.1038/nprot.2007.303.

Martinec, Ondrej, Martin Huliciak, Frantisek Staud, Filip Cecka, Ivan Vokral, and Lukas Cerveny. "Anti-HIV and Anti-Hepatitis C Virus Drugs Inhibit P-Glycoprotein Efflux Activity in Caco-2 Cells and Precision-Cut Rat and Human Intestinal Slices." Antimicrobial Agents and Chemotherapy 63, no. 11 (September 3, 2019): 7–9. https://doi.org/10.1128/AAC.00910-19.

Question 6: "those in in DAA-treated" in line 130 should revised.

Answer: corrected

Reviewer 2 Report

The authors have demonstrated the inhibitory effects of anti-HIV and anti-HCV drugs on the function of P-glycoprotein with different experimental systems (Caco-2 cells, human precision-cut intestinal slices).  The authors also compared their previous results demonstrating the inhibition potency of these drugs on P-gp with different substrate, Rhodamine 123 with the current results and found the inhibitory effect of drugs on P-gp is substrate-dependent.  Although the similar findings had already been proposed by previous reports, this reviewer feels that accumulation of such information is important for the future accurate evaluation of drug interaction risks.

Comments

1) Based on the current results, assuming that the mode of inhibition is competitive or non-competitive, apparent inhibition constant (Ki value) of each drug can be roughly estimated from the change in Papp ratio of digoxin in Caco-2 cells or cellular accumulation of digoxin in hPCIS (a complete inhibition of P-gp by CP100356 can be assumed.)  Then, [I1] (=Dose/250mL)/Ki value can be calculated as an indicator of the risk of DDIs at intestinal P-gp according to the regulatory guideline on DDIs.  Such table might be useful for the readers.

2) Has anyone compared the expression level of P-gp in Caco-2 cells and hPCIS?  The difference in the P-gp expression might affect the sensitive detection of the inhibitory effect of P-gp by drugs, which might lead to the different results of inhibition assay.

3) Anti-HIV and anti-HCV drugs are sometimes used as a combination drug.  The current data can tell us the relative contribution of each component to the overall P-gp inhibition.  Could the authors discuss the relative importance of each drug in P-gp inhibition for clinically-used combination drugs?

4) The authors found that rilpivirine and eltravirine can inhibit P-gp in Caco-2 cells, but not in hPCIS.  Since the enzymatic activity of Caco-2 cells is very low for some enzymes such as CYP3A, it is possible that rilpivirine and eltravirine are extensively metabolized in hPCIS and long-term exposure of these drugs could not be realized only in hPCIS.  Other possibility is that cellular uptake of rilpivirine and eltravirine is dominated by some uptake transporters and hPCIS decreased that function.  How have the authors considered such possibilities to explain the difference in the inhibition potency of P-gp in different experimental systems?

Author Response

Answers to the reviewer 2.

Dear reviewer,

Thank you for your valuable comments and questions. Our answers can be found below (and in the attached file). We hope you find our answers satisfactory.

Authors

Comments and Suggestions for Authors

Comment 1:

Based on the current results, assuming that the mode of inhibition is competitive or non-competitive, apparent inhibition constant (Ki value) of each drug can be roughly estimated from the change in Papp ratio of digoxin in Caco-2 cells or cellular accumulation of digoxin in hPCIS (a complete inhibition of P-gp by CP100356 can be assumed.) Then, [I1] (=Dose/250mL)/Ki value can be calculated as an indicator of the risk of DDIs at intestinal P-gp according to the regulatory guideline on DDIs. Such table might be useful for the readers.

Answer: Thank you for your suggestions. For the opponent's reference, we carried out a rough estimation of I1 values (see Table 1 and 2 below). However, we decided not to include this estimation in the revised manuscript. In our opinion, Ki and I1 values should be assessed in the hPCIS, i.e., an experimental model that better represents the real situation in the small intestine (Table 2). However, obtained human intestinal tissue samples are too small to perform extensive studies with more than three concentrations (at least four inhibitor concentrations for this purpose) needed for the precise calculation of Ki values (Burlingham et al., 2003; FDA, 2020). Additionally, we used digoxin as a probe at clinically irrelevant concentrations of 15 nM and 6 nM in hPCIS and Caco-2, respectively.

FDA. "In Vitro Drug Interaction Studies-Cytochrome P450 Enzyme-and Transporter-Mediated Drug Interactions Guidance for Industry," 2020. https://www.fda.gov/Drugs/GuidanceComplianceRegulatoryInformation/Guidances/default.htm.

Burlingham, Benjamin T., and Theodore S. Widlanski. “An Intuitive Look at the Relationship of Ki and IC50: A More General Use for the Dixon Plot.” Journal of Chemical Education 80, no. 2 (February 1, 2003): 214. https://doi.org/10.1021/ed080p214.

Table 1. Caco-2 – IC 50 and I1 calculations

Drug

Rough estimation of IC50 (µM)

Dose/250ml (µM)

I1

Asunaprevir

4.90

534.55

≥100

Atazanavir

12.92

1702.48

≥100

Daclatasvir

10.60

324.81

10 ≤ I1 ≤ 100

Darunavir

21.10

5842.97

≥100

Elbasvir

6.53

226.75

10 ≤ I1 ≤ 100

Etravirine

6.73

1837.88

≥100

Grazoprevir

6.92

521.57

10 ≤ I1 ≤ 100

Ledipasvir

56.23

404.94

≤10

Lopinavir

4.85

1272.24

≥100

Rilpivirine

1.02

272.91

≥100

Ritonavir

3.81

554.83

≥100

Saquinavir

8.60

2981.33

≥100

 Table 2. hPCIS - IC 50 and I1 calculations

Drug

Rough estimation of IC50 (µM)

Dose/250ml (µM)

I1

Asunaprevir

12.94

534.55

10 ≤ I1 ≤ 100

Atazanavir

31.89

1702.48

10 ≤ I1 ≤ 100

Daclatasvir

9.31

324.81

10 ≤ I1 ≤ 100

Darunavir

59.00

5842.97

10 ≤ I1 ≤ 100

Grazoprevir

39.58

521.57

10 ≤ I1 ≤ 100

Lopinavir

25.19

1272.24

10 ≤ I1 ≤ 100

Ritonavir

5.53

554.83

10 ≤ I1 ≤ 100

Saquinavir

13.14

2981.33

≥100

Velpatasvir

5.40

452.99

10 ≤ I1 ≤ 100

Comment 2:

Has anyone compared the expression level of P-gp in Caco-2 cells and hPCIS? The difference in the P-gp expression might affect the sensitive detection of the inhibitory effect of P-gp by drugs, which might lead to the different results of inhibition assay.

Answer: Direct comparison between hPCIS and Caco-2 cell line has not been done yet. We have performed literature search and found comparison of human jejunum with Caco-2 cells and we have added a paragraph dealing with this issue into the Discussion (273-281).

Comment 3:

Anti-HIV and anti-HCV drugs are sometimes used as a combination drug. The current data can tell us the relative contribution of each component to the overall P-gp inhibition. Could the authors discuss the relative importance of each drug in P-gp inhibition for clinically-used combination drugs?

Answer: Of the FDA-approved combination drugs for which we have tested each component, there is usually only one ABCB1 inhibiting compound, and thus the relative contribution of that compound to overall ABCB1 inhibition will be 100%. Examples include dolutegravir-rilpivirine, ledipasvir-sofosbuvir, or sofosbuvir-velpatasvir combinations, where we identified only one of the pair as an ABCB1 inhibitor. On the other hand, some regimens involving protease inhibitors use ritonavir as a pharmacokinetic booster that inhibits ABCB1 and CYP3A4, and at the same time, the other component of the combination also inhibits ABCB1 (HHS, 2022). In this case, estimating the relative contribution of the single drug to the overall ABCB1 inhibition is complicated. Examples of such combinations are lopinavir-ritonavir, darunavir-ritonavir, atazanavir-ritonavir. Although ritonavir came out as a more potent inhibitor in our study, it is used at a much lower dose compared to the other drug.

Moreover, we only tested the inhibitory potency of the antivirals on digoxin transport, so it is not easy to interpolate our data to other ABCB1 substrates. With certainty, we can only state which antivirals can influence the ABCB1-mediated transport of digoxin.

Panel on Antiretroviral Guidelines for Adults and Adolescents. Guidelines for the Use of Antiretroviral Agents in Adults and Adolescents with HIV. Department of Health and Human Services. Available at https://clinicalinfo.hiv.gov/sites/default/files/guidelines/documents/ AdultandAdolescentGL.pdf. Accessed on February 10th 2022

Comment 4: The authors found that rilpivirine and eltravirine can inhibit P-gp in Caco-2 cells, but not in hPCIS. Since the enzymatic activity of Caco-2 cells is very low for some enzymes such as CYP3A, it is possible that rilpivirine and eltravirine are extensively metabolized in hPCIS and long-term exposure of these drugs could not be realized only in hPCIS. Other possibility is that cellular uptake of rilpivirine and eltravirine is dominated by some uptake transporters and hPCIS decreased that function. How have the authors considered such possibilities to explain the difference in the inhibition potency of P-gp in different experimental systems?

Answer:  We agree with the reviewer that rilpivirine and etravirine could be metabolized in hPCIS during our experiments. Both drugs are substrates of CYP3A4 that is highly expressed in the human intestine, and therefore also in hPCIS, and almost missing in the Caco-2 cell line (Brück et al., 2017; Nakamura et al., 2002). Metabolism in hPCIS can therefore lead to the decreased intracellular concentration of the parent drug and thus decreased inhibition effect. We didn't measure the rate of the antivirals metabolism in the hPCIS, so we can't exclude this phenomenon. We discuss the effect of the metabolism on rilpivirine and etravirine inhibition on line 273.

Possible cellular uptake in hPCIS can also be considered as the reason for the observed results as there can be differences in expression compared to Caco-2 cell line. E.g. Caco-2 cell line expression for uptake transporters OATP2B1 is much higher compared to the jejunum tissue (Brück et al., 2017). Some antivirals were demonstrated to be substrates of the OATP family uptake transporters, but mostly OATP1A2 is mentioned in this case (Minuesa et al., 2011). As the OATP1A2 is lacking in the Caco-2 cell line and jejunum tissue, this will probably not be the reason for the observed differences. On the other hand, we can’t exclude other SLC family uptake transporters are involved in this process as differences between fresh tissue and Caco-2 cell line exist (Vaessen et al., 2017). We added this information also to the manuscript discussion part.

Brück, S., J. Strohmeier, D. Busch, M. Drozdzik, and S. Oswald. "Caco-2 Cells - Expression, Regulation and Function of Drug Transporters Compared with Human Jejunal Tissue." Biopharmaceutics & Drug Disposition 38, no. 2 (March 2017): 115–26. https://doi.org/10.1002/bdd.2025.

 Minuesa, Gerard, Isabel Huber-Ruano, Marçal Pastor-Anglada, Hermann Koepsell, Bonaventura Clotet, and Javier Martinez-Picado. "Drug Uptake Transporters in Antiretroviral Therapy." Pharmacology & Therapeutics 132, no. 3 (December 2011): 268–79. https://doi.org/10.1016/j.pharmthera.2011.06.007.

 Nakamura, Tsutomu, Toshiyuki Sakaeda, Nobuko Ohmoto, Takao Tamura, Nobuo Aoyama, Toshiro Shirakawa, Takashi Kamigaki, et al. "Real-Time Quantitative Polymerase Chain Reaction for MDR1, MRP1, MRP2, and CYP3A-MRNA Levels in Caco-2 Cell Lines, Human Duodenal Enterocytes, Normal Colorectal Tissues, and Colorectal Adenocarclnomas." Drug Metabolism and Disposition 30, no. 1 (2002): 4–6. https://doi.org/10.1124/dmd.30.1.4.

 Vaessen, Stefan F.C., Marola M.H. van Lipzig, Raymond H.H. Pieters, Cyrille A.M. Krul, Heleen M. Wortelboer, and Evita van de Steeg. “Regional Expression Levels of Drug Transporters and Metabolizing Enzymes along the Pig and Human Intestinal Tract and Comparison with Caco-2 Cells.” Drug Metabolism and Disposition 45, no. 4 (April 2017): 353–60. https://doi.org/10.1124/dmd.116.072231.
